# Ecophysiological Differentiation among Two Resurrection Ferns and Their Allopolyploid Derivative

**DOI:** 10.3390/plants12071529

**Published:** 2023-04-01

**Authors:** Luis G. Quintanilla, Ismael Aranda, María José Clemente-Moreno, Joan Pons-Perpinyà, Jorge Gago

**Affiliations:** 1School of Environmental Sciences and Technology (ESCET), University Rey Juan Carlos, 28922 Móstoles, Spain; 2National Institute for Agricultural and Food Research and Technology (INIA), Spanish National Research Council, 28040 Madrid, Spain; 3Agro-Environmental and Water Economics Institute (INAGEA), University of the Balearic Islands, 07122 Palma de Mallorca, Spain

**Keywords:** allopolyploidy, antioxidant activity, drought stress, elementome, fern, photosynthesis, leaf traits, ecological niche, water potential

## Abstract

Theoretically, the coexistence of diploids and related polyploids is constrained by reproductive and competitive mechanisms. Although niche differentiation can explain the commonly observed co-occurrence of cytotypes, the underlying ecophysiological differentiation among cytotypes has hardly been studied. We compared the leaf functional traits of the allotetraploid resurrection fern *Oeosporangium tinaei* (*HHPP*) and its diploid parents, *O. hispanicum* (*HH*) and O. *pteridioides* (*PP*), coexisting in the same location. Our experimental results showed that all three species can recover physiological status after severe leaf dehydration, which confirms their ‘resurrection’ ability. However, compared with *PP*, *HH* had much higher investment per unit area of light-capturing surface, lower carbon assimilation rate per unit mass for the same midday water potential, higher non-enzymatic antioxidant capacity, higher carbon content, and lower contents of nitrogen, phosphorus, and other macronutrients. These traits allow *HH* to live in microhabitats with less availability of water and nutrients (rock crevices) and to have a greater capacity for resurrection. The higher assimilation capacity and lower antioxidant capacity of *PP* explain its more humid and nutrient-rich microhabitats (shallow soils). *HHPP* traits were mostly intermediate between those of *HH* and *PP*, and they allow the allotetraploid to occupy the free niche space left by the diploids.

## 1. Introduction

Polyploids originate in low numbers within populations of diploid parents [1]. In theory, the initial establishment of polyploids and their subsequent coexistence with parents is constrained by a frequency-dependent mating disadvantage, known as minority cytotype exclusion [1]. Individuals of the less frequent cytotype are more likely to be fertilized by gametes from the more common cytotype, causing them to have odd-ploidy offspring, which are mostly sterile. Thus, the rarer cytotype may be progressively eliminated from admixed populations. Despite this initial disadvantage, polyploidization is a very frequent speciation mechanism in plants [2], and polyploids often form populations mixed with their diploid parents [3]. Explanations for this paradox include higher competitive ability of polyploids [4] and niche differentiation between cytotypes [5,6], which can increase the likelihood of coexistence. 

In allopolyploids, whole genome duplication occurs after an interspecific hybridization event. The coexistence of two divergent parental genomes in allopolyploid individuals results in profound genetic and epigenetic changes [7]. These, in turn, trigger shifts in responses to environmental factors, which may allow allopolyploids to diverge ecologically from parents [8] or outcompete them [9]. The phenotypic traits of allopolyploids, as in homoploid hybrids, can be intermediate between those of the parent species, biased toward or overlapping with one of the parents, or even be outside the variation range of parents (transgressive trait) [10,11]. This phenotypic diversity reflects the multiple expression patterns of duplicated (homoeologous) genes [12]. 

Despite great advances in the understanding of the genetic effects of allopolyploidization, its consequences on the expression of ecophysiological traits are still largely unknown [13,14]. Allopolyploids have long been viewed as ‘fill-in’ taxa that occupy geographic ranges and ecological niches intermediate to those of their diploid parents [15]. Some studies support that allopolyploids have intermediate abiotic niches that overlap those of their parents or maintain considerable niche conservatism [16,17]. However, two recent analyses indicate that niche divergence of allopolyploids from their parents is frequent [18,19]. Consistent with this, allopolyploids often have increased competitiveness in environments that are harsh or unsuitable for their parent diploids [20,21]. Fixed heterozygosity, intergenomic interactions, and gene expression dosage effects have been proposed to explain the better growth vigor [22,23] or better stress tolerance [24] observed in allopolyploids compared with their diploid ancestors [25].

Many allopolyploids display a better physiological capacity in dry environments than both diploid parents. These transgressive responses can be due not only to heterosis, but also to increase in cell size, or ‘gigas‘ effect, a nucleotypic consequence of duplicated genomes [26]. Cell enlargement in the xylem accompanied by smaller pit pore sizes could increase resistance to drought-induced hydraulic failure while maintaining significant hydraulic conductivity rates, which may explain why polyploids tend to occupy drier habitats that their parents [27]. In addition, compared with diploid parents, polyploids often have higher leaf thickness, higher leaf mass per area (LMA), and more pubescence [20,28], which are traits related to drought tolerance and efficient gas-exchange in arid and semi-arid environments [29]. Polyploids also tend to show larger guard cells with larger stomatal apertures, and lower stomatal density, which allow for greater water use efficiency (WUE) [21,24]. This maximizes carbon assimilation per water losses, a key factor for adaptation to xeric habitats [29].

Allopolyploidy also changes functional traits more closely linked to biochemistry. Compared to diploid parents, some polyploids show a delayed stomatal closure up to much lower water potentials, implying that photosynthesis can continue for a longer period driven by greater accumulation of osmolytes and osmoprotectors as proline and soluble sugars [20,28]. These processes are essential to finally obtain a net positive carbon balance at the end of the growing season, which ensures species viability in dry environments [30]. Research on the consequences of polyploidy on secondary metabolism has focused primarily on improving the production and chemical diversity of compounds in medicinal plants [31]. However, changes in the amount and, especially in allopolyploids, variety of compounds can affect both their competitive interactions with diploid parents and their tolerance to abiotic stress [23,32]. 

Interestingly, polyploidy is very common in vascular plants adapted to water scarcity by a ‘resurrection‘ strategy, such as many ferns [33,34]. Such resurrection plants are those that can tolerate dehydration in their vegetative tissues to almost complete depletion of water (<10% of the water content) for months [35,36]. As plants dry out, photosynthesis and all other metabolic activities slow until they cease or nearly so [35]. The mechanisms driving survival and preservation of functionality under these extreme desiccation events can basically be grouped into two different key-points: control of reactive oxygen species (ROS) levels by the antioxidant biochemistry, to avoid damage by oxidative stress during dehydration and rehydration, and structural rearrangements, which ensure cell integrity [34,37]. However, these structural features involved in survival to drastic dehydrations can constrain their photosynthetic capacity due to reduced mesophyll conductance (*g_m_*) associated with thicker cell walls [38,39], which implies longer optimal growing periods to fulfill a positive net carbon balance. 

Here, we compared the ecophysiology of three resurrection ferns: the allotetraploid *Oeosporangium tinaei* (Tod.) Fraser-Jenk. and its diploid parents, *O. hispanicum* (Mett.) Fraser-Jenk. & Pariyar and *O. pteridioides* (Reichard) Fraser-Jenk. & Pariyar. Hereafter, they will be referred to by genome constitution of the sporophyte: *HH* = *O. hispanicum*, *PP* = *O. pteidioides* and *HHPP* = *O. tinaei*, where *H* = *hispanicum* genome and *P* = *pteridioides* genome. These species belong to cheilanthoid clade (Pteridaceae: Cheilanthoideae), which is adapted to arid and seasonally dry environments. The study species curl up their leaves by dehydration during summer drought and uncurl them (leaf ‘resurrection’) when rain returns in autumn (Figure 1). *HHPP* and *HH* are mainly located in the Iberian Peninsula (SW Europe), while *PP* has a wider distribution in S Europe, N Africa and W Asia. *HHPP* exhibits substantial sympatry with both progenitors, which suggests that its niche is intermediate to those of its parents in terms of breadth and abiotic tolerances [17]. Indeed, in the localities where the three cytotypes coexist, there is some spatial segregation between them, which tend to form unmixed aggregates [40]. All three species inhabit siliceous rocks that are highly exposed to the sun, but *PP* grows on shallow soils, *HH* occupies crevices and concavities with hardly any soil, and *HHPP* is in intermediate microhabitats and sometimes grows in close proximity to one of the parents. 

Our main objective was to decipher the functional attributes responsible for this niche differentiation. For this, we studied co-occurring populations of the three species at the onset of summer drought, when they still had fully expanded leaves, but were beginning to undergo water stress. Specifically, we assessed the following leaf traits: leaf mass per area, maximum photosynthetic rate, midday water potential, desiccation tolerance, antioxidant capacity, and elemental composition. We hypothesize that *HHPP* traits respond to resource allocation trade-offs between carbon gains and stress tolerance in an intermediate manner relative to diploid progenitors, which drives niche intermediacy. 

## 2. Materials and Methods

### 2.1. Plant Species and Study Populations

The three studied *Oeosporangium* species are hemicriptophytes, with short rhizomes that may branch but do not have extensive clonal growth. Leaves are arranged in apical crowns and are longer in *HHPP* (up to 30 cm) and *HH* (<26 cm) than in *PP* (<18 cm) [41]. Laminae arebi- or tripinnate, with the underside covered with dense reddish long hairs in *HH* (Figure 1), short sparse hairs in *HHPP*, or glabrous in *PP*. All three species have a wide geographic distribution in Spain and are not included in the national Red List of threatened plants [42]. The study was carried out on one population of each species in Picadas dam (central Spain, 40°20′ N, 4°15′ W, altitude 530 m). The climate is continental Mediterranean, characterized by an extended summer drought, which subjects the vegetation to severe water stress and cold winters. Total annual precipitation is 437 mm, and mean annual temperature is 16.9 °C (years 2004−2022, Picadas dam weather station, Tagus Hydrographic Confederation). The study populations were located on sunny, SW oriented, rocky slopes, which consisted of schist and gneiss, and they had a few other plants, e.g., *Asplenium ceterach*, *Umbilicus rupestris,* and *Quercus ilex*. Field measurements and sample collections were conducted from 9 to 15 May 2022, when the summer drought was just beginning, and a few days before the three species curled all their leaves due to dehydration (own observations). For all the variables studied, the sample sizes (*n*) correspond to biological samples, i.e., individuals which were randomly selected. Some individuals were sampled repeatedly for several variables, depending on their leaf abundance. 

### 2.2. Leaf Structure and Water Content

One to two basal pinnae per plant (*n* = 15 individuals/species) were scanned and their area was measured with the program ImageJ [43]. Pinnae were then oven-dried for 48 h at 60 °C and weighed to determine their dry mass and then calculate LMA (g m^−2^). Relative water content (RWC; %) was used to determine the hydration status of leaves. One to two basal pinnae per plant (*n* = 7 individuals/species) were weighed to obtain the fresh weight (FW). Pinnae were subsequently hydrated in wet tissue paper for 24 h at 4 °C in darkness and weighed to obtain the turgid weight (TW). Finally, pinnae were dried (48 h, 60 °C) to obtain the dry weight (DW). RWC was calculated with the following equation: RWC = 100 × (FW − DW)/(TW − DW). 

### 2.3. Gas Exchange and Water Potential

Leaf gas exchange and water potential were measured in 9, 10, and 14 individuals of *HH*, *HHPP* and *PP*, respectively. Gas exchange and chlorophyll fluorescence measurements were performed, employing the equipment Li-6400XT (Li-Cor Inc., Lincoln, USA) with a fluorescence chamber of 2 cm^2^ (Li-6400-40). Leaves were carefully placed in the measurement chamber to ensure that they covered the maximum area and contacted the leaf thermocouple. If the leaf did not completely cover the chamber measurement area, a picture of that leaf was taken to recalculate the leaf area and its gas exchange parameters using ImageJ. Instantaneous measurements of light-saturated net photosynthesis (A_area_; μmol CO_2_ m^−2^ s^−1^) and stomatal conductance to H_2_O (g_s_; μmol H_2_O m^−2^ s^−1^) were recorded 20−30 min after clamping once leaves reached the gas-exchange steady-state. The set-up in the leaf chamber was: 400 ppm of C_a_ (atmospheric CO_2_ concentration in the measurement chamber), 1200 μmol m^−2^ s^−1^ of photosynthetic photon flux density (PPFD) (90:10% red:blue light), determined previously as light saturating conditions, 50–70% relative humidity, flow 150–300 μmol air s^−1^ (flow was reduced to maximize ΔCO_2_ when leaf gas-exchange rates were extremely low), and 25 °C block temperature. Mass-based net photosynthesis (A_mass_; μmol CO_2_ g^−1^ s^−1^)was obtained by dividing A_area_ (μmol CO_2_ m^−2^ s^−1^) by LMA (g m^−2^). Quantum yield of the photosystem II (Φ_PSII_; unitless) was calculated as described in [44]. Leaf water potential at mid-day (Ψ_md_; MPa) was measured with a Scholander chamber (PMS 1505D-EXP, PMS Instrument Company, Albany, USA) on the plants just after recording gas exchange. In addition, leaves were reserved in paper bags and oven-dried until reaching stable dry weight for subsequent analysis of elemental composition. 

### 2.4. Desiccation Tolerance

To assess leaf desiccation tolerance, the so-called ‘Falcon test‘ [45] was performed. This test consists of four steps as follows. (i) Hydration: pinnae separated from the leaf were fully hydrated in wet tissue paper for 24 h to obtain TW and initial maximum photochemical efficiency of PSII (F_v_/F_m_; unitless), measured with a Junior-Pam (Walz, Germany). (ii) Desiccation: these pinnae were subsequently desiccated for 24 h in closed 50-mL Falcon tubes with a sponge soaked in three desiccants: NaCl, MgCl_2_, and silica gel, which promote atmospheres with relative humidities of 80, 50, and 10%, respectively. (iii) Recovery: pinnae were fully rehydrated as in (i). (iv) Complete drying: pinnae were oven-dried at 60 °C for 48 h to obtain their DW. After (ii) and (iii), pinnae were weighed to calculate RWC after desiccation (RWC_desic._) and recovery (RWC_recov._), respectively. F_v_/F_m_ was also measured after (ii) and (iii). F_v_/F_m(desic.)_ was the ratio between F_v_/F_m_ after desiccation and initial F_v_/F_m_, whereas F_v_/F_m(recov.)_ was the ratio between F_v_/F_m_ after recovery and initial F_v_/F_m_. The variable F_v_/F_m(recov.)_ is a proxy of desiccation tolerance [45]. For this Falcon test, *n* = 17 individuals/species, from each of which one to two pinnae were used. 

### 2.5. Antioxidant Capacity and Elemental Composition

Total antioxidant capacity was measured in 11, 16, and 16 individuals of *HH*, *HHPP,* and *PP*, respectively. One to two basal pinnae per individual were harvested and immediately frozen in liquid nitrogen. Frozen samples were lyophilized and subsequently ground and homogenized with acidified methanol (7% acetic in 80% methanol) buffer. Trolox Equivalent Antioxidant capacity (TEAC) was calculated by ABTS radical scavenging capacity assay employing a Trolox standard curve [46]. Results are expressed as mM Trolox equivalent (TE) per g of DW. 

To study foliar elemental composition, two to three basal pinnae per plant (*n* = 7 individuals/species) were harvested and separated in two subsamples. One subsample was subsequently oven-dried at 80 °C and ground to obtain a dry powder for quantifying total carbon (C_total_; g/100 g DW), nitrogen (N_total_; g/100 g DW), and other elements. The second subsample was used to obtain cell wall (CW) fraction (as AIR: alcohol insoluble residue) as follow: samples were boiled in absolute ethanol until they were bleached. Subsequently, samples were cleaned in acetone by shaking for 30 min twice, further air-dried overnight, and homogenized by dry milling. These CW samples were used to determine C and N content specifically in cell walls (C_CW_ and N_CW_, respectively; g/100 g of AIR). These two variables could finally be determined in six of the seven sampled individuals of each species. C and N determinations (C_total_, N_total_, C_CW_ and N_CW_) were obtained by combustion at 950 °C and analyzed by individual infrared detection and thermal conductivity. Total content of other elements was determined in an ICP THERMO ICAP 6500 DUO spectrometer (Thermo Scientific, Rockford, USA). Both analyses were performed by the CEBAS-CSIC Ionomic Service in Murcia, Spain.

### 2.6. Statistical Analyses

LMA, RWC, A_area_, TEAC, C_total_, N_total_, C/N_total_, C_CW_, N_CW_, C/N_CW_, and 13 additional foliar elements were analyzed by one-way ANOVA, with species as the fixed factor. RWC_desic._, F_v_/F_m(desic.)_, RWC_recov._, and F_v_/F_m(recov.)_ were analyzed by a two-way ANOVA, with species and desiccation level as fixed factors. For improving normality, these four dependent variables and RWC were arcsine-transformed before the analyses. Subsequent pairwise comparisons were made using Tukey tests (*p* < 0.05). One-way ANCOVAs were conducted to compare A_mass_ among the three species (species = categorical predictor variable). Three models were performed, each including the following covariates: Ψ_md_, g_s_, and Φ_PSII_. The models were first run with the interaction categorical variable × covariate, and then the non-significant interactions were excluded. Post hoc comparison of adjusted least square means of species were performed with Bonferroni multiple testing procedure. Principal component analysis (PCA) was carried out to compare the foliar elemental composition among species. This PCA included C_total_, N_total_ and the 13 additional elements with more variability explained by the first two principal components, based on vector length. All statistical analyses were conducted in R software [47]. Results are given as means ± standard error unless otherwise indicated. The complete dataset is available in the Appendix A.

## 3. Results

### 3.1. Leaf Structure and Water Content 

LMA of the three species were significantly different (*F*_2, 42_ = 25.00, *p* < 0.0001, ANOVA; *p* < 0.05, all Tukey tests). LMA of *HH* (186 ± 10 g m^−2^) was almost twice that of *PP* (97 ± 3 g m^−2^), and *HHPP* showed an intermediate LMA (140 ± 11 g m^−2^). RWC showed no difference among *PP* (88.3 ± 1.5%), *HHPP* (88.3 ± 1.6%), and *HH* (82.0 ± 2.8%) (*F*_2, 18_ = 2.93, *p* = 0.0793, ANOVA). The range for RWC of these species were 82−93%, 82−94%, and 68−92%, respectively.

### 3.2. Gas Exchange and Water Potential

A_area_ of *HH*, *HHPP,* and *PP* were 4.88 ± 0.82, 4.26 ± 1.01, and 4.02 ± 0.73 μmol CO_2_ m^−2^ s^−1^, respectively, with no significant differences among them (*F*_2, 30_ = 0.26, *p* = 0.77, ANOVA). As expected, A_mass_ related positively with Ψ_md_, g_s_, and Φ_PSII_ (Table 1, Figure 2). The dependence of A_mass_ on these three covariates was parallel for all species (i.e., the species × covariate interaction was not significant). Moreover, the Y intercepts of the regression lines were significantly different among species for both Ψ_md_ and g_s_, i.e., the species means while controlling for these covariates were different. It should be noted that the operating range of Ψ_md_ was lower (more negative) in *PP* than in *HH* (Figure 2a). When adjusting for this covariate, *PP* had ten times higher A_mass_ than *HH* (0.060 vs. 0.006 μmol CO_2_ g^−1^ s^−1^), whereas, when adjusting for g_s_ (Figure 2b), *PP* showed three times higher A_mass_ than *HH* (0.046 vs. 0.014 μmol CO_2_ g^−1^ s^−1^). *HHPP* had intermediate means, more similar to that of *HH* after adjusting for Ψ_md_, or closer to that of *PP* after adjusting for g_s_ (*p* < 0.05, Bonferroni test). After controlling for Φ_PSII_, differences in A_mass_ among species were marginally significant in ANOVA (*p* = 0.053; Table 1) or not significant in the post hoc Bonferroni test (*p* > 0.05 for all species pairs; Figure 2c). 

### 3.3. Desiccation Tolerance

RWC_desic._ did not differ significantly among species but did differ among desiccation levels (Appendix A). RWC_desic._ at 80% and 50% RH desiccations were similar (means ~18%, data of the three species pooled; Appendix A), and significantly higher than that at 10% desiccation (3.5 ± 1.0%). F_v_/F_m(desic.)_ showed significant differences both among species and among desiccation levels (Appendix A, Appendix A). F_v_/F_m(desic.)_ for the species decreased in the order: *PP* (49.3 ± 4.0%) > *HHPP* (36.8 ± 4.0%) > *HH* (33.1 ± 3.2%), and means for the desiccations decreased: 50% RH (49.0 ± 4.0%) > 10% RH (40.2 ± 3.7%) > 80% RH (32.1 ± 3.7%). By contrast, RWC_recov._ did not differ significantly among desiccations (Table 2, Figure 3). RWC_recov._ was significantly higher in *PP* (77.5 ± 2.9%) than in *HHPP* and *HH* (means ~63%). F_v_/F_m(recov.)_ did not show significant differences among species or among desiccation levels (Table 2, Figure 3). The overall mean of this variable for all species−desiccation combinations (*n* = 48) was 75.8 ± 1.8%. 

### 3.4. Antioxidant Capacity and Elemental Composition 

Leaf TEAC differed significantly among species (*F*_2, 40_ = 7.09, *p* = 0.0023, ANOVA). Specifically, *HHPP* (94.7 ± 6.3 mM TE/g DW) and *HH* (92.6 ± 8.6 mM TE/g DW) showed similar antioxidant activities (*p* > 0.05, Tukey test), which were significantly higher than that of *PP* (66.2 ± 4.1 mM TE/g DW) (*p* < 0.05, Tukey test). 

Species also differed significantly in the leaf variables C_total_ (*F*_2, 17_ = 48.9, *p* < 0.0001, ANOVA), N_total_ (*F*_2, 17_ = 11.49, *p* = 0.0007) and C/N_total_ (*F*_2, 17_ = 15.45, *p* = 0.0002). *HH* had more C_total_ and less N_total_ than *PP*, and, thus, C/N_total_ was higher in *HH* (Table 3). N_total_ and C/N_total_ of *HHPP* were more similar to those of *HH*, whereas C_total_ of *HHPP* was similar to that of *PP*. In the cell wall, C and N contents were much more similar among species, which did not show significant differences in N_CW_ (*F*_2, 15_ = 0.62, *p* = 0.55) or C/N_CW_ (*F*_2, 15_ = 0.64, *p* = 0.54). In the three species, C/N_CW_ had much lower values than C/N_total_. C_CW_ did have significant differences among species (*F*_2, 15_ = 7.10, *p* = 0.0068) and, similar to C_total_, it was higher in *HH* than in *PP* (Table 3). C_CW_ of *HHPP* was intermediate relative to diploids.

Most of the 13 additional leaf elements analyzed differed significantly among species (Appendix A). Al, Ca, Fe, K, Mg, Mo, P, and S were more abundant in *PP* than in *HH*, while Cd and Mn were more abundant in *HH*. *HHPP* had intermediate contents compared with diploids for many elements. In the PCA (Figure 4), the first principal component (PC1) explained 54% of the data variance and PC2 accounted for 17% of variance. *HH* and *PP* were located in the negative and positive values of PC1, respectively, while *HHPP* again showed an intermediate position between diploids. The highest negative loading of PC1 was related to C, Mn, and Cd, whereas the positive side was mainly driven by macronutrients as N, P, K, and S (Figure 4).

## 4. Discussion

Our comparison of leaf functional traits between the allotetraploid resurrection fern *HHPP* and its diploid parents, *HH* and *PP*, yields two key findings. First, *HH* and *PP* have diverged for most of the traits assessed, despite phylogenic proximity, as they are congeneric, and despite living sympatrically in the same rocky environment. Specifically, *HH* showed leaf traits linked to drier conditions than those of *PP*, such as much higher LMA, lower A_mass_, higher TEAC, higher C_total_, C/N_total_ and C_CW_, and lower N_total_. Second, we found no evidence of transgressive trait expression in the allotetraploid, as its traits fell within the variation range of parents and were intermediate in most cases. These results support our hypothesis that *HHPP* expresses mainly intermediate traits, which can favor this polyploid to successfully fill an intermediate niche between those of diploids. 

LMA of *HH*, *HHPP,* and *PP* (186, 140 and 97 g m^−2^, respectively) are in the range of other rupicolous resurrection ferns, which show larger LMA, leaf thickness and A_area_ values [39] than desiccation sensitive ferns [48]. For example, sun-exposed individuals of *Asplenium ceterach* have LMA = 123 g m^−2^, while shade individuals show much lower LMA due to both lower leaf thickness and lower leaf density [49]. This species is a good reference for comparison, as it is present in the vicinity of the populations we studied. *Anemia caffrorum*, another resurrection rupicolous fern, produces desiccation-tolerant leaves in the dry season, with the same LMA as *PP*, and desiccation-sensitive leaves in the wet season, with half the LMA [50]. A_area_ of *HH*, *HHPP,* and *PP* (4.9, 4.3 and 4.0 μmol CO_2_ m^−2^ s^−1^, respectively) were lower than those of other rupicolous resurrection ferns [39,49,50,51]. However, direct comparisons should be made with caution, as the individuals in our study were beginning to suffer from summer water deficiency. This is evidenced by the variation in Ψ_md_ and g_s_ (Figure 2), which strongly affect CO_2_ assimilation rates [52,53]. This variation in water status provided an exceptional opportunity to compare the dehydration responses of the three species. After considering the impact of those covariates on carbon uptake (Figure 2), adjusted mean A_mass_ of *PP* was similar to those other rupicolous resurrection ferns, such as *A. ceterach* [49]. Interestingly, *PP* was able to maintain a positive net carbon gain at much more negative Ψ_md_ (−2.2 MPa) than *HH* (−1.2 MPa), whereas *HHPP* showed an intermediate capacity. Moreover, at any g_s_, *HH* had lower carbon assimilation than *HHPP* and especially *PP*, which indicates that *HH* has stronger diffusive mesophyll limitations, i.e., lower g_m_, probably driven by leaf anatomy combined with higher biochemical limitations [38]. In general, assimilation capacity (A_mass_) is not only negatively associated with investment per unit area of light-capturing surface (i.e., LMA), but also negatively correlated with both leaf longevity [54] and stress tolerance [55]. Thus, higher LMA and lower A_mass_ of *HH* compared with *HHPP,* and especially with *PP*, indicate that the leaves of the former are more durable and more stress tolerant. These results fit well with the leaf resurrection capacity of these species, which is higher in *HH* than in *PP* and intermediate in *HHPP* (own unpublished data).

We also experimentally compared the leaf dehydration tolerance among species. In the desiccation stage, all three species underwent dramatic water losses, especially at 10% RH desiccation (RWC_desic._ = 3.5%, Appendix A). The same stress tolerance test has been carried out on seven other resurrection ferns and seven resurrection angiosperms [45]. After identical desiccation treatments, their water content declines were less severe than those of our study species. In the recovery stage (24-h rehydration) of our experiment, the three species achieved good rehydration, although it was higher in *PP* than in *HHPP* and *HH*, regardless of the level of previous desiccation (Figure 3a). This result could indicate that leaves of *HHPP* and *HH* need more time to regain their turgor, which may be due to denser tissues and/or thicker cell walls [37]. We found no significant differences in the recovery of photosynthetic status among species or among desiccation levels, with an overall 76% F_v_/F_m(recov.)_ for all species–desiccation combinations (Figure 3c). This high physiological recovery is similar to those of other resurrection ferns, which in turn are better than those of resurrection angiosperms [45]. High F_v_/F_m(recov.)_ thus confirms the outstanding adaptation of the three species to tolerate extreme tissue dehydrations. These results and carbon assimilation rates in response to leaf water potentials (see above) indicate that, over a wide range of water statuses, the allopolyploid does not have a better photosynthetic performance than both diploid parents. In other words, *HHPP* showed no transgressive assimilation capacity. Similarly, leaf functional traits of several allopolyploid wood ferns (*Dryopteris*) were mostly intermediate between those of diploid parents [11]; however, these ferns are desiccation sensitive. A well studied model of plant resurrection is the hexaploid angiosperm *Ramonda serbica* and the related diploid *R*. *nathaliae*. As our study species, they both coexist and hybridize [56], and thus there may be competitive or reproductive exclusions [1] among them. Their recovery of F_v_/F_m_ during leaf rehydration is high and similar between species, as in our desiccation experiment [36]. In addition, the diploid shows higher net carbon uptake, less sensitive to high temperatures, and other xeromorphic traits allowing it to occupy drier habitats compared to the hexaploid [36]. Thus, this polyploid did not display a better photosynthetic capacity in dry conditions, as *HHPP* compared with *PP*. 

Leaf TEAC was very high in the three ferns studied compared to the few values available for other plant species, e.g., [57,58]. As with other resurrections plants, the three ferns are homoiochlorophyllous (own observations), i.e., retain chlorophylls and thylakoidal organization during the dry stage [59]. These chlorophylls will continue to absorb light after dehydration, and the energy not transferred to the electron transport chain will trigger the overproduction of ROS [60]. Consequently, these ferns were expected to have such a strong antioxidative response, which is an important photoprotection strategy in desiccation-tolerant plants [34,59]. Consistent with this, *PP*, which has a lower resurrection capacity than *HHPP* and *HH*, also showed a lower TEAC. Reduced investment in secondary metabolites with antioxidant capacity allows more resources to be devoted to photobiochemistry (Rubisco and electron transport chain) [55,61], which can contribute to the higher photosynthetic performance of *PP*.

Foliar elemental composition was also very different between the diploids and again intermediate in *HHPP* for most of the elements. The higher N_total_ in *PP* may also explain its higher photosynthetic capacity, since most of the nitrogen is accumulated in the chloroplast (up to 75%), with Rubisco accounting for ca. 50% of the photosynthetic N [61,62]. Contents of other macronutrients (Ca, K, Mg, P, and S) were also higher in *PP* than in *HH*, which is explained by the fact that *PP* grows on soils with presumably higher nutrient availability than the rock crevices where *HH* settles. According to the biochemical niche hypothesis, the elementome of a species defines its biochemical niche [63], which is expected to be more different between phylogenetically distant species and between sympatric species to reduce competitive pressure [64]. Our results indicate that the three species studied, despite their phylogenetic proximity, occupy distinct biogeochemical niches, intermediate in *HHPP*, which allows them to coexist. Higher C and lower N and P concentrations, together with slower growth rate, again indicate that *HH* has a more stress-tolerant strategy than *HHPP* and *PP* [63].

CW compounds are major components of leaf dry mass, especially in plants with high LMA [55]. The study species, in addition to high LMA, are expected to have very thick CW, as this is common in other cheilanthoid resurrection ferns [39]. Compared with *PP*, *HH* had more C_CW_, which may be due to a higher investment in hemicelluloses and pectins, rich in C, and lacking in N. Both compounds lead to elastic and flexible CW to sustain cell integrity under extreme tissue desiccation events [37]. Other protective mechanisms, related to the higher resurrection capacity and tolerance to stress of *HH*, such as the accumulation of sugars and the production of antioxidant metabolites and protective pigments, may explain its higher C/N_total_ [65]. Related to this, the abundant reddish hairs of *HH* lamina protect the homoiochlorophyllous chloroplasts from excess light during dehydration [36,59], as leaves expose their underside when curling (Figure 1).

## 5. Conclusions

The current study found that sympatric populations of two diploid resurrection ferns, and their allotetraploid derivative have great ecophysiological differentiation, which promotes their coexistence by niche divergence. All three species can recover photosynthetic function (F_v_/F_m(recov.)_) after extreme desiccation events. However, one of the diploids (*HH*) has more ‘conservative’ leaves [66], with much higher investment per unit area of light-capturing surface (LMA), lower carbon assimilation rate per unit mass (A_mass_), higher antioxidant capacity (TEAC), higher C content (C_total_ and C_CW_), and lower contents of N, P, and other macronutrients. This combination of leaf traits allows it to live in microhabitats with less availability of water and nutrients (rock crevices) and to have a greater capacity for resurrection. The other diploid (*PP*) shows more ‘acquisitive’ leaves [66], with enhanced assimilation capacity and lower ability to cope with oxidative stress. These leaf traits allow it to occupy more humid and nutrient-rich microhabitats (shallow soils). Trait divergence and the resulting niche segregation between both diploid parents leave a large free adaptive space that has been filled by the allotetraploid (*HHPP*). Its intermediate abiotic niche is well explained by a combination of leaf traits driving A_mass_ and dehydration tolerance in an intermediate manner between those of the parent diploids. Our findings suggest that these different ecophysiological traits drive niche differences, reducing interspecific competition and thus increase the capacity of each species to recover from low density, which is critical for stable coexistence [67]. Our study also shows how useful diploid–polyploid contact zones are for understanding the evolutionary success of whole genome duplication. 

## Figures and Tables

**Figure 1 plants-12-01529-f001:**
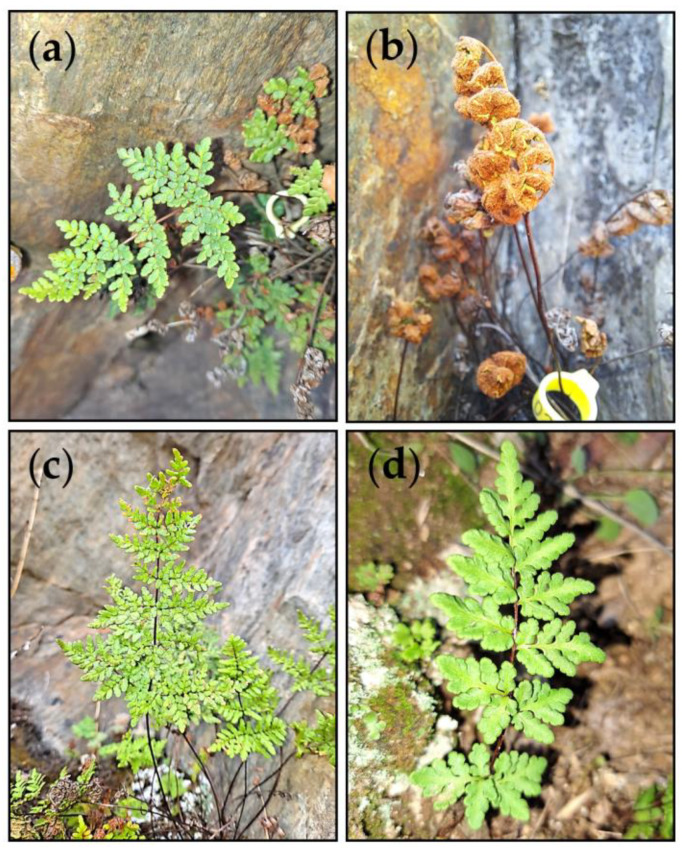
Leaves of the studied Oesporangium species: (**a**) O. hispanicum (HH), well hydrated; (**b**) HH, completely dry; (**c**) O. tinaei (HHPP), well hydrated; (**d**) O. pteridioides (PP), well hydrated.

**Figure 2 plants-12-01529-f002:**
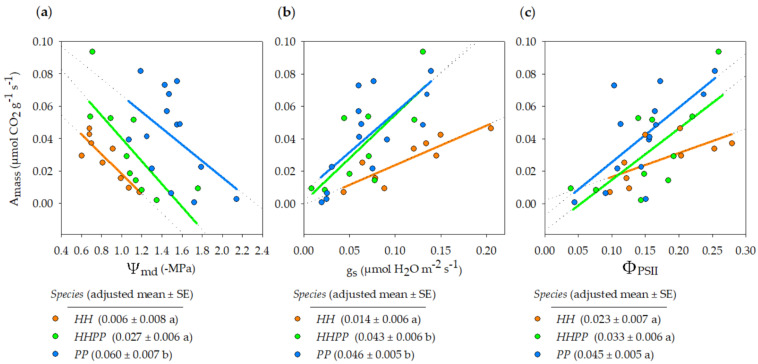
Variation of A_mass_ among the three *Oesporangium* species while controlling for three covariates: (**a**) Ψ_md_; (**b**) g_s_; (**c**) Φ_PSII_. In the tables below each figure, different letters indicate significantly different means (*p* < 0.05, Bonferroni tests). *n* = 9−14 individuals per species. See Table 1 for ANCOVA results.

**Figure 3 plants-12-01529-f003:**
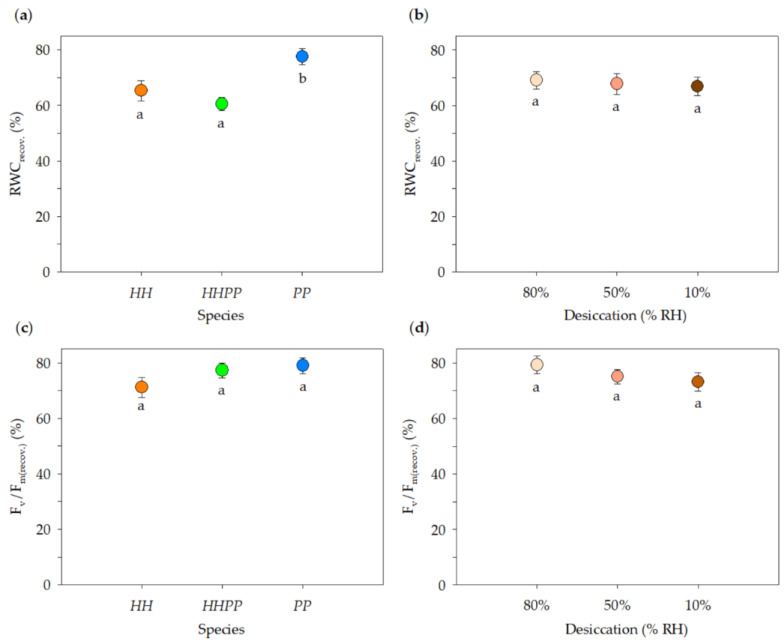
Mean values (± SE) of RWC_recov._ and F_v_/F_m(recov.)_ for the three *Oesporangium* species after three desiccation levels (80%, 50%, and 10% relative humidities) followed by rehydration (recovery): (**a**) RWC_recov._, comparison of species; (**b**) RWC_recov._, comparison of desiccation levels; (**c**) F_v_/F_m(recov.)_, comparison of species; (**d**) F_v_/F_m(recov.)_, comparison of desiccation levels. Different letters indicate significantly different means (*p* < 0.05, Tukey tests). *n* = 17 individuals per species. See Table 2 for ANOVA results.

**Figure 4 plants-12-01529-f004:**
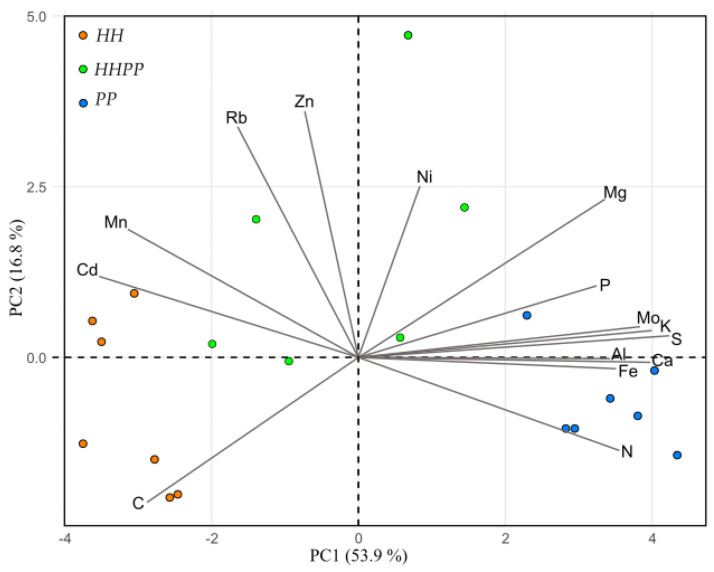
Principal component analysis (PCA) biplot of the foliar elemental contents (variables) in the individuals (samples) of the three *Oeosporangium* species. PC = principal component. *n* = 6−7 individuals per species.

**Table 1 plants-12-01529-t001:** ANCOVAs for testing differences in mass-based net photosynthesis (A_mass_) among the three *Oesporangium* species. Three covariates were included in separate models: leaf water potential at mid-day (Ψ_md_), stomatal conductance (g_s_), and quantum yield of the photosystem II (Φ_PSII_). The species × covariate interaction was not significant (*p* > 0.05) in any of the models, so they were repeated without this interaction. Significant effects (*p* < 0.05) are indicated in bold. Sample size (*n*) was 9, 10, and 14 individuals of *HH*, *HHPP,* and *PP*, respectively.

Source of Variation	df ^1^	SS	*F*	*p*
Species	2	0.008	9.91	**0.0005**
Ψ_md_	1	0.008	19.80	**0.0001**
Residual	29	0.011		
Species	2	0.005	8.10	**0.0020**
g_s_	1	0.009	29.5	**<0.0001**
Residual	28	0.009		
Species	2	0.002	3.26	0.0530
Φ_PSII_	1	0.008	21.44	**<0.0001**
Residual	29	0.011		

^1^ df = degrees of freedom; SS = sum of squares.

**Table 2 plants-12-01529-t002:** ANOVAs for testing differences in relative water content (RWC) and maximum photochemical efficiency of PSII (F_v_/F_m_) among the three *Oesporangium* species after three desiccation levels (80%, 50%, and 10% relative humidities) followed by rehydration (recovery). F_v_/F_m(recov.)_ is the ratio (F_v_/F_m_ after recovery): (initial F_v_/F_m_). Significant differences (*p* < 0.05) are indicated in bold. *n* = 17 individuals per species.

Variable	Source of Variation	df ^1^	SS	*F*	*p*
RWC_recov._	Species	2	1289.7	8.13	**0.0011**
	Desiccation	2	16.8	0.11	0.8997
	Sp. × Desic.	4	89.0	0.28	0.8889
	Residual	41	3252.8		
F_v_/F_m(recov.)_	Species	2	228.3	1.64	0.2075
	Desiccation	2	175.4	1.26	0.2956
	Sp. × Desic.	4	307.9	1.10	0.3683
	Residual	39	2718.5		

^1^ df = degrees of freedom; SS = sum of squares.

**Table 3 plants-12-01529-t003:** Mean values (±SE) of total and cell wall (CW) carbon, nitrogen, and carbon/nitrogen ratio in leaves of the three *Oesporangium* species. Units of C_total_ and N_total_ are g/100g of dry weight (DW), whereas C_CW_ and N_CW_ are in g/100g of alcohol insoluble residue (AIR). Different letters indicate significantly different means (*p* < 0.05, Tukey tests). *n* = 6−7 individuals per species.

		Species	
Variable	*HH*	*HHPP*	*PP*
C_total_	51.4 ± 0.3 a	47.9 ± 0.2 b	48.7 ± 0.2 b
N_total_	1.77 ± 0.06 a	1.86 ± 0.07 a	2.27 ± 0.10 b
C/N_total_	29.2 ± 1.0 a	26.0 ± 1.1 a	21.7 ± 0.9 b
C_CW_	45.6 ± 0.2 a	44.6 ± 0.6 a,b	43.3 ± 0.4 b
N_CW_	2.83 ± 0.16 a	3.12 ± 0.22 a	2.80 ± 0.28 a
C/N_CW_	16.4 ± 1.1 a	14.7 ± 1.0 a	16.2 ± 1.4 a

## Data Availability

The raw data are included in the Appendix A.

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
