# Peer review of "Ecophysiological Differentiation among Two Resurrection Ferns and Their Allopolyploid Derivative"

_plants, 2023, doi:10.3390/plants12071529_

Round 1
Reviewer 1 Report
In the submitted manuscript, Quintanilla et al. compared the leaf functional traits of the allotetraploid resurrection fern HHPP, HH and PP, and showed that all three species have ‘resurrection’ ability, recovering physiological status after severe leaf dehydration, and also have differences: HH had much higher investment per unit area of light-capturing surface, etc., PP had the higher assimilation capacity and lower antioxidant capacity, while HHPP exhibited the intermediate type. This study uncovers the ecophysiological differentiation among cytotypes. Here are my detailed comments.
1. The authors should show the replicated number for these experiments in Tables 1, 2, and 3, Figures 2 and 3, the same as for the Supplemental Figures and Tables.
2. The citation format of the article should be revised according to the requirements of the Journal.
3. Line229, Page7‘2.7. Statistical Analyses’should change to‘2.6. Statistical Analyses’.
Author Response
We are grateful for your helpful suggestions (in italics below), which we have incorporated into the revised manuscript.
- The authors should show the replicated number for these experiments in Tables 1, 2, and 3, Figures 2 and 3, the same as for the Supplemental Figures and Tables.
All tables and figures now have sample sizes.
- The citation format of the article should be revised according to the requirements of the Journal.
We have adapted the references to the journal format.
- Line229, Page7‘2.7. Statistical Analyses’should change to‘2.6. Statistical Analyses’.
This has been corrected in line 223 of the revised version.
Reviewer 2 Report
I read with great pleasure the manuscript, which deals with a very interesting subject and is very well written.
I have only a few minor comments. First of all, please pay attention to the author guidelines of the journal and place numbers for citations in the text.
More information on the occurrence of the analysed species, such as their frequency in the Iberian Peninsula and their conservation status, would be interesting.
Also some additional comments on the population studied: firstly, how many populations of each species were analysed, and are there differences between the populations of the three species in terms of substrate and plant communities?
Please justify why the analyses were carried out only in May and not repeated during the summer, when drought is more intense.
Author Response
We are grateful for your helpful suggestions (in italics below), which we have incorporated into the revised manuscript.
First of all, please pay attention to the author guidelines of the journal and place numbers for citations in the text.
We have adapted the references to the journal format.
More information on the occurrence of the analysed species, such as their frequency in the Iberian Peninsula and their conservation status, would be interesting.
We have added this information in lines 136-138 of the revised manuscript.
Also some additional comments on the population studied: firstly, how many populations of each species were analysed, and are there differences between the populations of the three species in terms of substrate and plant communities?
We have added this information in lines 138 and 144-145 of the revised manuscript.
Please justify why the analyses were carried out only in May and not repeated during the summer, when drought is more intense.
We have clarified that the leaves of the studied populations curled due to dehydration a few days after our study (lines 147-148). We have also added in the Introduction (lines 91-92) that dehydrated plants interrupt their functions, to make it more evident that our gas exchange variables could not be determined in this quiescent state.